# Flowering Response of *Cannabis sativa* L. 'Suver Haze' under Varying Daylength-Extension Light Intensities and Durations

Jongseok Park [1], Cristian E. Collado [2], Vu Phong Lam [1] and Ricardo Hernández [2,*]

1 Department of Horticultural Science, Chungnam National University, Daejeon 34134, Republic of Korea
2 Department of Horticultural Science, North Carolina State University, Raleigh, NC 27695, USA
* Correspondence: ricardo_hernandez@ncsu.edu or rhernan4@ncsu.edu

**Abstract:** Daylength-extension lighting (DE) is used in the cannabis industry to increase plant size and produce cuttings by regulating flowering and extending the vegetative stage. Growers have reported incomplete or transitional inflorescences in several *Cannabis* cultivars even when exposed to long photoperiods. *Cannabis sativa* L. 'Suver Haze' has been reported to develop incomplete inflorescences in North Carolina nurseries using photoperiods of 15 h. The objectives of this study were to investigate the required light intensity and photoperiod to inhibit the flowering of 'Suver Haze'. In Experiment 1, DE of 1.0, 2.5, 5.8, and 10.3 $\mu mol \cdot m^{-2} \cdot s^{-1}$ of photosynthetic photon flux density from incandescent lamps were used to extend the photoperiod of 'Suver Haze' from 9 to 15 h. A 9 h photoperiod control was included. The results showed that all DE treatments stopped the full transition to flowering compared to the control; however, all DE-treated plants showed the presence of incomplete inflorescences. In Experiment 2, three photoperiod treatments of 15 h, 18 h, and 21 h were tested. 'Suver Haze' under 18 h and 21 h photoperiods did not develop incomplete inflorescences in contrast to plants in 15 h photoperiod. Therefore, a light intensity of at least 1.0 $\mu mol \cdot m^{-2} \cdot s^{-1}$ PPFD and an 18 h photoperiod are required to prevent incomplete inflorescences and flowering of 'Suver Haze'.

**Keywords:** hemp; photoperiod; cannabis flower; day-extension; cannabis cuttings

## 1. Introduction

With the changing legal status of *Cannabis sativa* production, the market is expanding in Europe, Oceania, and North America [1]. Cannabis has been widely cultivated due to its industrial [2], ornamental [3], nutritional [4], medicinal, and recreational [5] potentials. From regulatory and application perspectives, cannabis plants are categorized based on the level of Δ9-tetrahydrocannabinol (THC), one of the most important phytocannabinoids [6]. Plants are generally classified and regulated as industrial hemp if it contains less than 0.3% THC in the dried flower (this level varies by country) or drug-type with more than this threshold [7]. In the United States, the cultivation of *Cannabis sativa* with less than 0.3% of THC is legal at the federal level [8], and the recreational use of high-THC plants continues to be legalized at the state level [9]. The current variability of plants from seeds leads to high variability in flowering initiation [10,11]. Currently, the majority of *Cannabis sativa* cultivated for medicinal and recreational markets is vegetatively propagated from stock-plant cuttings. Commercial greenhouse nurseries providing rooted cuttings to the field sometimes rely on natural or day-extension lighting to keep plants vegetative. However, growers have observed that stock plants of some cultivars develop incomplete inflorescence (Figure 1) while still maintaining overall vegetative growth. Most of the cannabis commercial materials are considered quantitative short day plants [12,13]. However, some day-length sensitive plants do not flower with photoperiods longer than 13 to 14 h, while others need 18 h or more to stop flowering [14]. In the United States (U.S.) a common CBD *Cannabis sativa* cultivar 'Suver Haze' shows a partial transition to flowering in the nursery

even under 15 h photoperiods. This partial transition is identified by the growers as the development of incomplete inflorescences on all node positions of the plant (Figure 1), which is accompanied by a reduction in the rate of stem extension and reduced apical dominance. Even though this response is more likely attributed to photoperiod effects, it is important to also consider the adequate intensity to trigger day-length extension responses since this can vary between plant species [15].

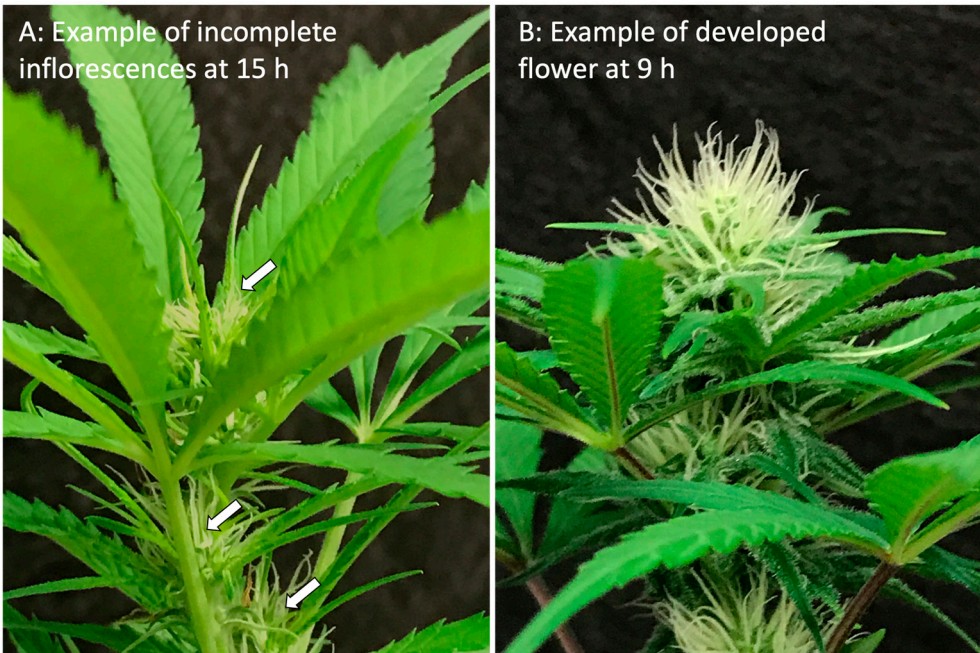

**Figure 1.** Incomplete or rudimentary inflorescent (**A**) and fully developing inflorescent (**B**) from the top of the canopy of two plants (*Cannabis sativa* L., 'Suver Haze') at the same age. (**A**) was grown under a 15 h photoperiod and (**B**) under a 9 h photoperiod for 28 days. Both plants were grown under the same daily light integral. Notice the incompletely developed flowers (arrows) and longer internode lengths on (**A**) compared to (**B**).

The objective of this study was to investigate the minimum light intensity and the required photoperiod to prevent incomplete inflorescence development. Two experiments were conducted: (1) to elucidate the required light intensity for daylength-extension (DE); and (2) to identify the long-day photoperiod to prevent full and partial transitions to flowering. Two hypotheses were tested: (1) low light intensity (1–10 $\mu$mol m$^{-2}$ s$^{-1}$ photosynthetic photon flux density (PPFD)) is sufficient to prevent the transition from vegetative growth to flowering of 'Suver Haze' (Experiment 1); and (2) 'Suver Haze' requires a photoperiod longer than 15 h to fully prevent the transition from vegetative growth to flowering (Experiment 2).

## 2. Materials and Methods

### 2.1. Plant Material and Growth Conditions

*Cannabis sativa* stem cuttings were taken from vegetative stock plants of 'Suver Haze' (© Oregon CBD). The stem cuttings were planted in a 72-cell tray (plant density 483 plants m$^{-2}$) for rooting in a peat-based horticultural medium (Fafard 4P Mix; Sun Gro Horticulture Co., Ltd., Agawam, MA, USA). During rooting, plants were maintained in a growth chamber under an air temperature of 26 °C (average day temperature), a relative humidity of 85–95%, and 120 $\mu$mol·m$^{-2}$·s$^{-1}$ PPFD with a 16 h photoperiod for two weeks. The cuttings were irrigated with tap water once a day.

### 2.2. Plant Growing Conditions

Rooted cuttings were transplanted to 1-gallon pots filled with sand medium and placed inside a growth chamber (EGC, reach-in chamber, Environmental Growth Chambers, Chagrin Falls, OH, USA). Plants were irrigated as needed with a complete hydroponic solution with an electrical conductivity of 1.0 ds m$^{-1}$ and a pH of 6.0. The nutrient solution contained (mg L$^{-1}$): 84 (N), 7.8 (P), 121 (K), 45/7 (Ca), 9.54 (Mg), 11.2 (S), 3.6 (Fe), 0.05 (Mn), 0.21 (Zn), 0.09 (Cu), 0.19 (B). The size of the growth chambers was 1.2 m in width, 0.9 m in depth, and 1.2 m in height, and each was outfitted with a combination of cool-white fluorescent and incandescent fixtures (Figure 2). For Experiment 1, the plant density inside the chamber was 4 plants per square meter for a total of 20 plants (4 plants per treatment/chamber) per replication. For Experiment 2, the final plant density was 6 plants per meter square for total of 18 plants (6 plants per treatment/chamber).

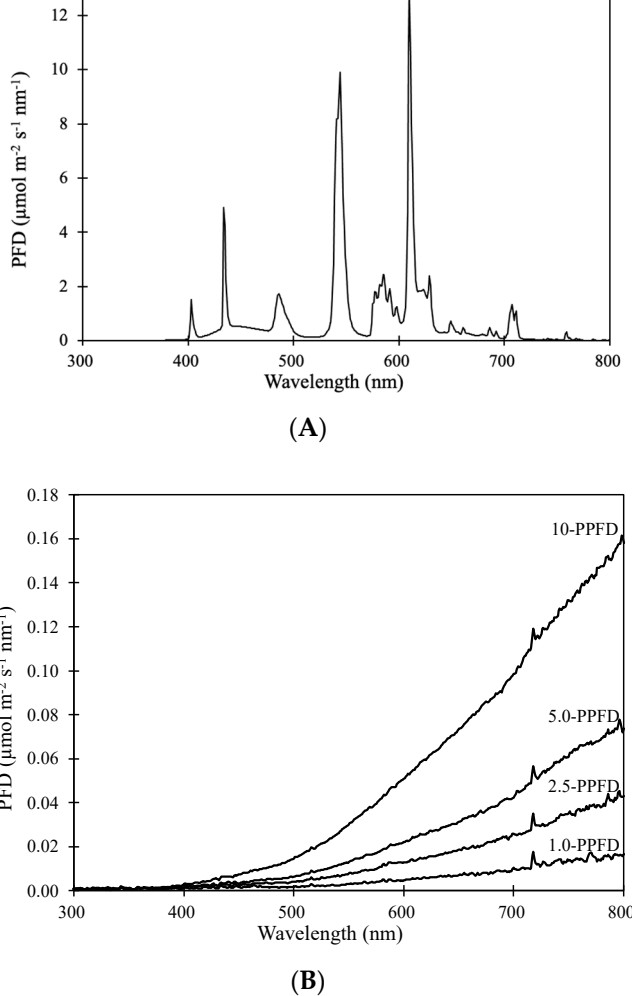

**Figure 2.** (**A**) represents the spectrum (cool-white-fluorescent fixture, T5 54 W) used for plant growth under a photoperiod of 9 h (experiments 1 and 2). (**B**) represents the daylength-extension spectrum (40 W incandescent fixtures) for four light intensity treatments of 1.0, 2.5, 5.0, 10.0 μmol·m$^{-2}$·s$^{-1}$ photosynthetic photon flux density. Detailed values (mean ± SD) for PPFD (400–700 nm) and total photon flux (300–800 nm) are presented in Table 1.

**Table 1.** Environmental conditions measured in the growth chambers (mean ± standard deviation) in the two experiments. Light measurements were collected at the beginning and end of each repetition, temperature and relative humidity were recorded every minute. In Experiment 1, five daylength-extension lighting (DE) treatments were tested (two replications in time). In Experiment 2, three different photoperiods were tested once.

| Experiment 1 [a] | DE-0 | DE-1.0 | DE-2.5 | DE-5.0 | DE-10 |
|---|---|---|---|---|---|
| DE-PPFD (400–700 nm: $\mu mol \cdot m^{-2} \cdot s^{-1}$) | 0 | $1.0 \pm 0.3$ | $2.5 \pm 0.5$ | $5.8 \pm 1.2$ | $10.3 \pm 1.7$ |
| DE-FR (700–800 nm: $\mu mol \cdot m^{-2} \cdot s^{-1}$) | 0 | $1.2 \pm 0.3$ | $3.0 \pm 0.5$ | $7.1 \pm 1.2$ | $12.0 \pm 1.7$ |
| DE-PF (300–800 nm: $\mu mol \cdot m^{-2} \cdot s^{-1}$) | 0 | $2.2 \pm 0.6$ | $5.5 \pm 1.1$ | $12.9 \pm 2.8$ | $22.2 \pm 3.6$ |
| DE-R:FR | 0 | 0.524 | 0.540 | 0.524 | 0.564 |
| PPFD ($\mu mol \cdot m^{-2} \cdot s^{-1}$) | | | $400.6 \pm 2.6$ | | |
| DE-Photoperiod (h) | 0 | 6 | 6 | 6 | 6 |
| PPFD-Photoperiod (h) | | | 9 | | |
| Photoperiod (h) | 9 | 15 | 15 | 15 | 15 |
| Daily light integral ($mol \cdot m^{-2} \cdot d^{-1}$) | | | $13.1 \pm 0.1$ | | |
| Temperature (°C) | | | $24.5 \pm 0.3$ | | |
| Relative humidity (%) | | | $69.8 \pm 6.9$ | | |
| **Experiment 2 [a]** | **15 h** | **18 h** | **21 h** | | |
| Photoperiod (h) | 15 | 18 | 21 | | |
| DE-Photoperiod (h) | 6 | 9 | 12 | | |
| PPFD-Photoperiod (h) | | 9 | | | |
| DE-PPFD (400–700 nm: $\mu mol \cdot m^{-2} \cdot s^{-1}$) | | $9.0 \pm 0.2$ | | | |
| DE-FR (700–800 nm: $\mu mol \cdot m^{-2} \cdot s^{-1}$) | | $10.5 \pm 0.2$ | | | |
| PPFD[y] ($\mu mol \cdot m^{-2} \cdot s^{-1}$) | | $463.0 \pm 5.0$ | | | |
| Daily temp (°C) | | $25.5 \pm 0.3$ | | | |
| Relative humidity (%) | | $63.6 \pm 2.2$ | | | |

[a] Incandescent lamps were used for the DE, and cool-white-fluorescent lamps were used to provide PPFD light intensity (Figure 2).

### 2.3. Lighting

For both experiments, fluorescent fixtures and incandescent lamps were installed at the top of the chamber (1.2 m height) to provide light for growth and daylength extension, respectively (Figure 2). Nine light measurements per chamber were taken using a spectroradiometer (PS-200, Apogee Instruments, Logan, UT, USA), and their average per treatment are presented in Table 1 and Figure 2. Plants were placed inside the chamber on a height adjustable platform. The platform height was adjusted (lowered) twice a week to adjust for increase in plant height and maintain the same PPFD and DE light intensities at the apical meristem of the plant by maintaining the same distance between the plant and the light fixtures. To achieve the different DE intensities, the 40W incandescent bulbs were covered with different layers of black shad-cloth. A diagram depicting the chamber set up is presented (Figure S1).

### 2.4. Treatments

In Experiment 1, all the plants were exposed to a similar PPFD of $400.6 \pm 2.6$ µmol·m⁻²·s⁻¹ for 9 h. In addition, plants were exposed for 6 h daily to five DE treatments of 0, 1.0, 2.5, 5.0, and 10 µmol·m⁻²·s⁻¹ PPFD for a total of 28 days (Table 1 and Figure 2). One treatment per chamber was used.

In Experiment 2, plants were exposed to three photoperiodic treatments (one treatment per chamber) with the same 9 h at $463.0 \pm 5.0$ PPFD and different hours of DE. The three photoperiods were: (1) 15 h (9 h PPFD and 6 h DE), (2) 18 h (9 h PPFD and 9 h DE, and (3) 21 h (9 h PPFD and 12 h DE) (Table 1). The dark periods were 9, 6, and 3 h, respectively, for a 24 h day. Experiment 2 was conducted for 24 days and only conducted one time.

### 2.5. Measurement of Plant Growth Parameters

For Experiment 1, the number of visible flowers, stem length, and number of nodes were counted at 0, 7, 14, 21 and 28 days after transplanting. Stem length was measured from the base of the stem to the apical meristem and the final number of nodes was counted from the fifth bottom leaf to the apical meristem. Shoot and root fresh mass were measured using a micro weighing scale (CAS MW-II; CAS Co., Ltd., East Rutherford, NJ, USA) at day 28. Shoot fresh mass was partitioned into stem fresh mass and leaf fresh mass. Shoot and root samples were placed in a drying oven at 70 °C for 7 days and dry mass was determined using a micro-weighing scale (CAS MW-II).

For Experiment 2, the presence or absence of incomplete inflorescences was recorded at end of the experiment (24 days).

### 2.6. Statistical Analysis

For Experiment 1, four plants (subsamples) per chamber (experimental unit) and two replications in time were used for analysis. For each replication, the chambers were randomly assigned to the treatments. The number of flowers, stem length, and number of nodes (Figure 3) were analyzed on days 0, 7, 14, 21, and 28. The treatment effect on fresh and dry mass was analyzed for roots, stem, and canopy (leaves and flowers) using ANOVA and Tukey's HSD mean separation. All analyses were performed using JMP software 14.2 (SAS Institute, Cary, NC, USA).

For Experiment 2, six plants were assigned per chamber/treatment. Since the treatment effect on the absence/presence of incomplete inflorescences was clearly recognizable in every plant, it was conducted only one time. The photoperiod inflorescence results from the experiments 1 and 2 were analyzed in an incomplete block design using generalized regression with Poisson distribution and Lasso estimation method (lower AICc) in JMP Pro 17.0 from SAS.

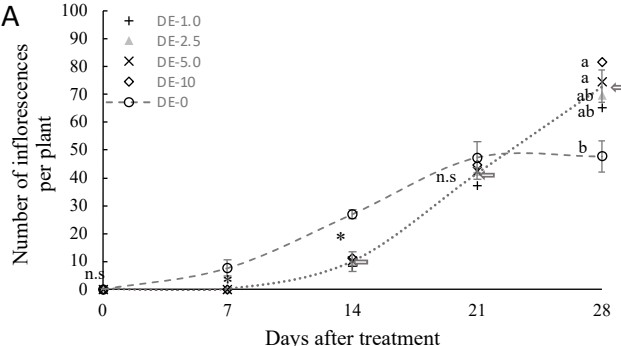

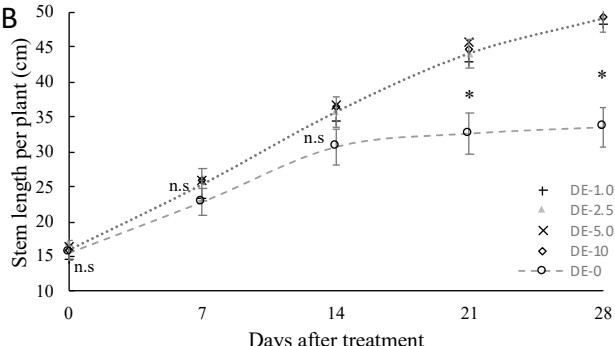

**Figure 3.** *Cont.*

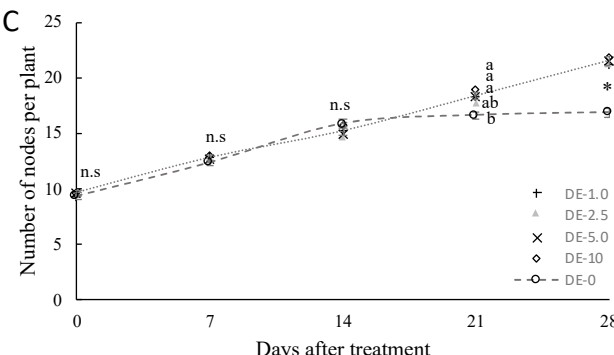

**Figure 3.** Effect of daylength-extension lighting with intensities of 0 (DE-0), 1 (DE-1.0), 2.5 (DE-2.5), 5 (DE-5.0), and 10 (DE-10) $\mu mol \cdot m^{-2} \cdot s^{-1}$ PPFD (400–700 nm) on the number of inflorescences (**A**), stem length (**B**), and number of nodes per plant (**C**) of *Cannabis sativa* 'Suver Haze' grown in a growth chamber for 28 days after treatment. Arrows on (**A**) indicate the development of incomplete inflorescences (see Figure 1). Data are presented as the mean ± SE. Different letters/asterisk show significant differences between treatments on each day, n.s shows no significant differences (Tukey's test, $\alpha = 0.05$). The DE treatments >0 $\mu mol \cdot m^{-2} \cdot s^{-1}$ extended the photoperiod from 9 to 15 h and showed no significant differences between them in (**A–C**); therefore, only one mean error bar from those treatments is presented per day.

## 3. Results and Discussion

### 3.1. Experiment 1: Daylength-Extension Experiment

3.1.1. Effect of Light Intensity on Flower Transition

Even though all plants under DE-1.0, DE-2.5, DE-5.0, and DE-10 showed the development of incomplete inflorescences (Figure 1), the 15 h photoperiod in all treatments was able to prevent the full flowering transition of 'Suver Haze' from the vegetative stage to the reproductive stage (Figures 3A and 4) and associated effects (Figure 3B,C). In addition, no differences were detected between 1 to 10 $\mu mol \cdot m^{-2} \cdot s^{-1}$. Figure 3A shows earlier flower initiations in the control treatment (DE-0) on days 7 and 14 (a higher number of inflorescences than all other DE treatments), reaching the maximum number of inflorescences by day 21. After Day 21, the biomass of existing flowers continued to increase in the control treatment. In the 1 to 10 $\mu mol \cdot m^{-2} \cdot s^{-1}$ DE treatments (DE-1.0, DE-2.5, DE-5.0, and DE-10), incomplete inflorescences started developing by day 14 and continued to develop until the end of data collection (Day 28, Figures 1 and 3A); these flowers did not increase in biomass and size (data not shown). In addition, plants in 1 to 10 $\mu mol \cdot m^{-2} \cdot s^{-1}$ DE treatments also continued to increase stem length and number of nodes when compared to the control (Figure 3B,C, respectively). This indicates that plants were still in vegetative development. On day 14, plants under DE-1.0, DE-2.5, DE-5.0, and DE-10 started showing the development of incomplete inflorescences, and apical dominance (stem height, node development) was not reduced compared to the control (DE-0), suggesting that a uniform light intensity of 1 $\mu mol \cdot m^{-2} \cdot s^{-1}$ PPFD (1.2 $\mu mol \cdot m^{-2} \cdot s^{-1}$ PF 700–800 nm; 2.2 $\mu mol \cdot m^{-2} \cdot s^{-1}$ PF 300–800 nm, Table 1) is sufficient to trigger photoperiodic responses and prevent a full transitioning to flowering.

Research reports have shown that plants can perceive very low light intensities. For example, Whitman et al. [16] showed that *Coreopsis verticillate* "Moonbeam" (long-day plant) was able to detect light intensities as low as 0.05 $\mu mol \cdot m^{-2} \cdot s^{-1}$. Most short-day plants can effectively perceive low light intensities to prevent flowering either as DE lighting or night interruption lighting. In addition, short day plants can also detect light intensity from a variety of spectra in both the PAR and far-red wavelengths [15]. The results of the present study are in agreement with known photoperiodic responses of ornamental plants [15,17]. Therefore, for 'Suver Haze', a uniform light intensity for day extension of at least 1 $\mu mol \cdot m^{-2} \cdot s^{-}$ (lowest tested in this experiment) is recommended to prevent

flowering. Therefore, this study is in agreement with recent published reports in other *Cannabis* cultivars [12,13].

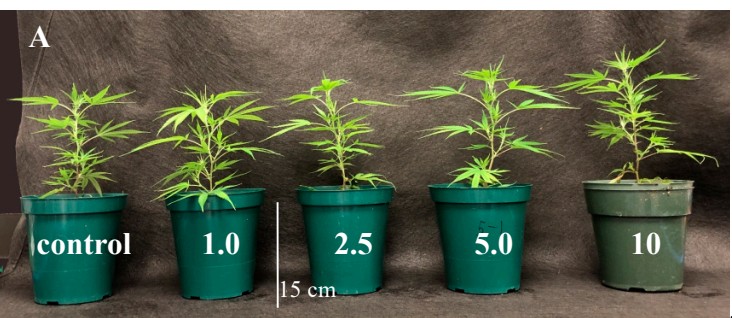

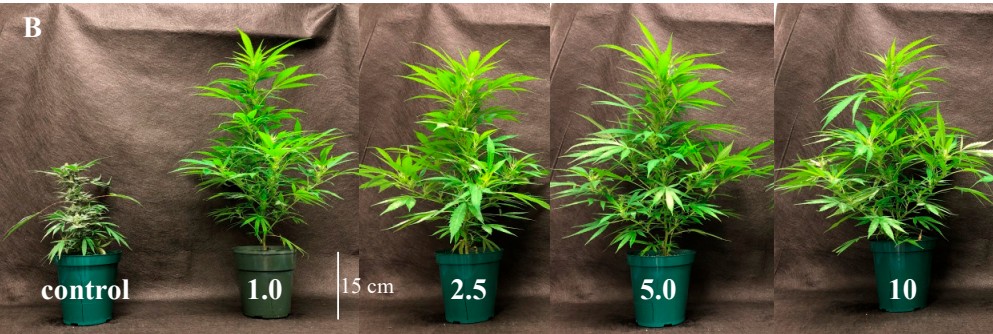

**Figure 4.** *Cannabis sativa* 'Suver Haze' under daylength-extension lighting treatments of 0, 1.0, 2.5, 5.0, and 10 $\mu$mol·m$^{-2}$·s$^{-1}$ PPFD (400–700 nm), respectively. Plants at treatment start (Day 0) (**A**) and 28 days after treatment (**B**).

### 3.1.2. Incomplete Inflorescent Development

In the present study, the flowering under the control treatment (9 h photoperiod) started at day 7, whereas incomplete inflorescent development (Figures 1 and 3A) was present in all DE treatments starting at day 14. This incomplete inflorescent development is known as the onset of the transition to the flowering stage (pers comm. J. Faust). Incomplete inflorescence development, during long-day conditions, is undesirable since flowers do not fully develop leading to a decrease in flower quality and yield. Since all the 15 h DE treatments had similar development of incomplete inflorescences, it was concluded that insufficient light intensity was not the cause of the development of incomplete inflorescences.

Other environmental stimuli can also trigger the transition to flowering. For example, the temperature during the vegetative stage is known to affect the flowering capacity of cannabis plants [18–21]; research on fiber hemp cultivars has shown that the optimal temperature for growth is 29 °C, and 306 to 636 °C degree days (base temperature of 1 °C) are required for the completion of the thermal requirements to initiate the day-length (photoperiod) dependent stage [19]. In the present study, vegetative cuttings from mature cloned stock plants were rooted and grown under average temperatures of 26 °C during the rooting stage (14 days) and 24.5 °C during the first week of the experiment with a total of 535 degree days. Moreover, under greenhouse non-supplemental lighting conditions, 'Suver Haze' cuttings in misting systems show flowers with only three weeks after sticking for rooting; while the visualization of flowers takes between 2 and 3 weeks for 'Suver Haze' plants after photoperiod is reduced from 18 h, or longer, to 12 h (Collado and Hernández, unpublished data). Therefore, it is unlikely the temperature in the present study influenced any unexpected flowering responses. Other stressors can also trigger the flowering transition, including root restriction (root-bound), water stress, and nutrient stress [22–24]. In the present experiment, the pot size was adequate and fertigation was properly managed during the experiment, hence these restrictions are unlikely to occur.

A more plausible explanation for the development of incomplete inflorescences in the present experiment is that the long-day critical photoperiod to fully prevent flowering for 'Suver Haze" is longer than 15 h. Therefore, Experiment 2 investigated the impact of different photoperiods on flower initiation.

### 3.1.3. Growth and Morphology

As expected, plants under the DE treatments had different morphological characteristics than those under the 9 h photoperiod control, including longer and heavier stems, greater number of nodes, and root mass (Figures 3C, 4 and 5). However, leaf and flower biomass (fresh mass and dry mass) were similar in all treatments and the control. After transitioning to flowering, short-day plants change photo-assimilate partitioning to favor developing flowers; consequently, other organs, such as leaves, nodes, stems, and roots have a reduction of growth rate and development [25]. This is consistent with the present results. The similarity in total flower mass between the short and longer photoperiod plants is likely caused by a maximum yield potential based on fewer sites for flower induction (i.e., nodes) in plants under 9 h photoperiod, while more nodes should have compensated the smaller inflorescences under 15 h conditions. Therefore, additional research focused on the optimal plant size (number of primary and secondary branches, leaves, etc.) for the optimal transition point to the reproductive stage is needed to maximize flower yield. Currently, the industry relies on the "number of weeks" as the quantitative measurement to change the photoperiod and initiate flowering. However, environmental conditions and agronomic practices greatly affect plant size, making the "number of weeks" prediction very inconsistent between growing systems.

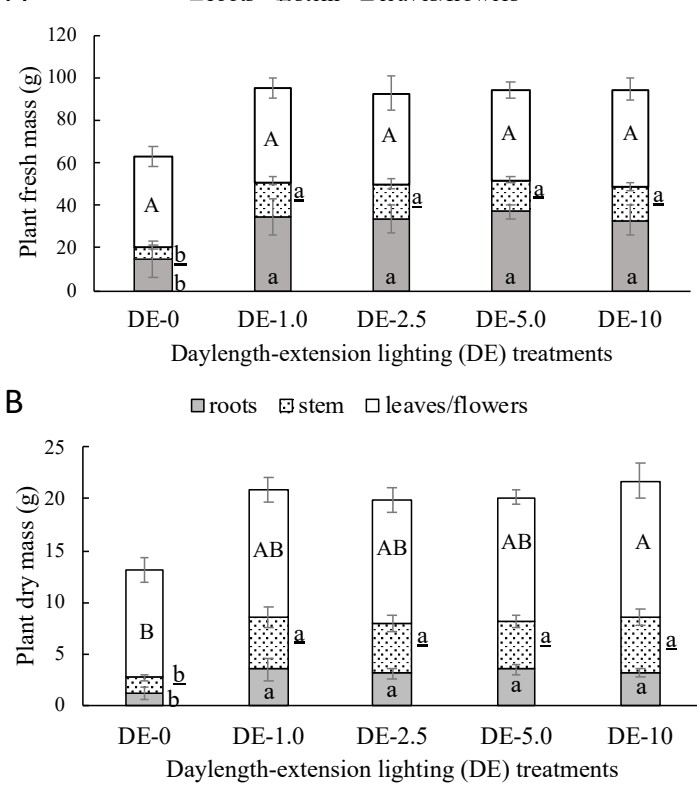

**Figure 5.** Effect of daylength-extension lighting (DE) with intensities of 0 (DE-0), 1.0 (DE-1.0), 2.5 (DE-2.5), 5.0 (DE-5.0), and 10 (DE-10) $\mu mol \cdot m^{-2} \cdot s^{-1}$ PPFD (400–700 nm) on shoot fresh mass (**A**), and shoot dry mass (**B**) of *Cannabis sativa* 'Suver Haze' grown in a growth chamber for 28 days. Data are presented as the mean $\pm$ SE. Different letters show significant differences between treatments (Tukey's test, $\alpha = 0.05$).

### 3.2. Experiment 2: Photoperiod

To investigate the photoperiod to completely prevent the development of inflorescence in "Suver Haze", plants were exposed to three long-day photoperiods: (1) 15 h (9 h PPFD and 6h DE), (2) 18 h (9 h PPFD and 9 h DE, and 3) 21 h (9 h PPFD and 12 h DE) using incandescent lamps (Figure 2, Table 1). All plants under the 15 h photoperiod treatment developed incomplete inflorescences (Figure 6) consistent with the previous experiment (Figures 1 and 4). In contrast, all plants under the 18 h and 21 h photoperiod treatments did not develop incomplete inflorescences ($\alpha$ = 0.05). Based on the previous DE experiment results, such as the presence of incomplete inflorescences and greater stem lengths, plants in 15 h treatment (Experiment 2) had partial transitioning to flowering. Therefore, the photoperiod to prevent flowering of 'Suver Haze' is greater than 15 h. In this experiment, 18 h was sufficient to prevent any incomplete inflorescences development.

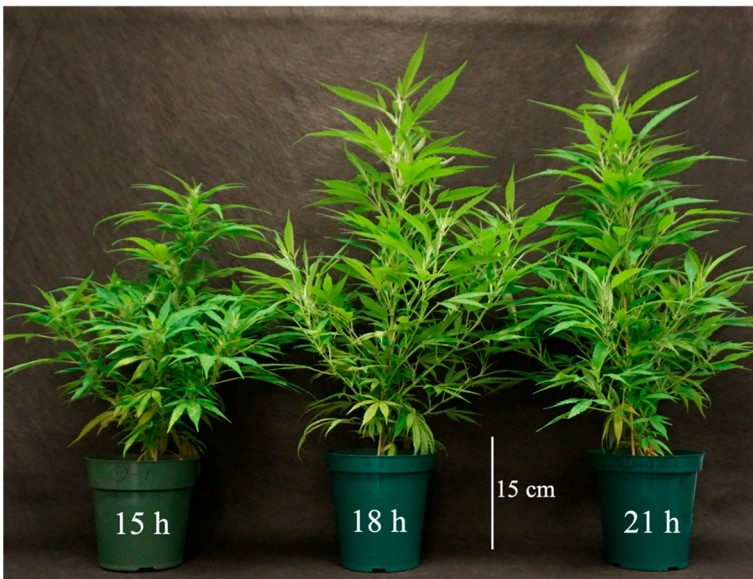

**Figure 6.** *Cannabis sativa* 'Suver Haze' under different photoperiod treatments composed of 15 hour (15 h), 18 hour (18 h), and 21 hour (21 h). Note the development of incomplete inflorescences under the 15 h treatment.

*Cannabis sativa* is considered a quantitative short-day plant and at least 10.8 h of an uninterrupted dark period is required to induce flowering [13,23]. However, research has documented high variability in the critical short-day photoperiod for flowering in *Cannabis sativa* [13,14,20,23,24]. For example, for field-grown *Cannabis sativa*, it has been reported that genotypes from different geographical origins have different critical photoperiods for flowering [26–28] ranging from 11 to 15 h. A recent study investigated the impact of photoperiod on flowering using a tissue culture system and found that 12–13.2 h was the critical photoperiod to initiate flowering, while 14.4 h or greater was suitable for preventing flowering [23]. In another study, Zhang et al. [24] found that the critical photoperiod to initiate flowering varied among 27 cultivars and within cultivars from different locations. This highlights the importance of understanding the photoperiodic responses of specific genotypes. Zhang et al. [24] also reported that small changes in the photoperiod (15 min increments) can be enough to affect the transition to flowering. In general, all the cultivars studied (except for day neutrals) had a critical photoperiod of 15 h or less to initiate flowering.

Several research studies have focused on the critical photoperiod to initiate flowering; however, less research reports are available on the adequate photoperiod to fully prevent flowering [13,29,30]. Such information is important for the nursery industry, which seeks to maximize the vegetative output (branching). In the present study, 'Suver Haze' required more than 15 h of light to prevent flowering, and even though additional research is

needed to find the exact critical photoperiod, 18 h of light was adequate to prevent the development of any incomplete inflorescences. Therefore, with the current information, our photoperiodic recommendation to prevent incomplete inflorescent formation in 'Suver Haze' is to use an 18 h photoperiod.

## 4. Conclusions

A photoperiod of 15 h and a daylength-extension lighting of 1 $\mu mol \cdot m^{-2} \cdot s^{-1}$ PPFD (1.2 $\mu mol \cdot m^{-2} \cdot s^{-1}$ PF 700–800 nm; 2.2 $\mu mol \cdot m^{-2} \cdot s^{-1}$ PF 300–800 nm; lowest intensity tested in this experiment) maintained plants vegetative but with production of incomplete inflorescences; however, a photoperiod of 18 h and a DE light intensity of 1 $\mu mol \cdot m^{-2} \cdot s^{-1}$ prevented 'Suver Haze' from flowering. Increasing the daylength-extension levels from 1 to 10 $\mu mol \cdot m^{-2} \cdot s^{-1}$ PPFD did not reduced the growth of incomplete inflorescences. In the current study, the DE lighting was provided using incandescent fixtures which are rich in far-red light. However, research has shown that the inhibition of flowering in short-day plants can be accomplished with most electric lamps [15]. Therefore, LEDs can be a more efficient lighting technology since the inhibition of flowering of short-day plants is less influenced by spectrum [15]. However, the effects of different light spectrums under extreme low light levels have not been studied yet for 'Suver Haze' or other short-day cultivars.

Additional research is required to further understand and characterize (1) the impact of light pollution (less than 1 $\mu mol \cdot m^{-2} \cdot s^{-1}$) on flower crops and (2) the critical photoperiods to prevent incomplete inflorescences for common cannabis cultivars in the US. In addition, efforts must be made in new studies to use the same genotypes or cultivars from the same sources.

**Supplementary Materials:** The following supporting information can be downloaded at: https://www.mdpi.com/article/10.3390/horticulturae9050526/s1, Figure S1. Schematic of chamber set up.

**Author Contributions:** Conceptualization, J.P., C.E.C. and R.H.; methodology, J.P.; software, J.P., V.P.L.; validation, J.P., C.E.C., V.P.L. and R.H.; formal analysis, C.E.C. and R.H.; investigation, J.P., C.E.C.; resources, J.P., C.E.C., V.P.L. and R.H.; data curation, J.P., C.E.C., V.P.L. and R.H.; writing—original draft preparation, J.P. and V.P.L.; writing—review and editing, C.E.C. and R.H.; visualization, J.P., C.E.C., V.P.L. and R.H.; supervision, R.H.; project administration, R.H.; funding acquisition, J.P. and R.H. All authors have read and agreed to the published version of the manuscript.

**Funding:** This research was funded by the Cooperative Research Program for Agriculture Science and Technology Development grant number [PJ017063022023], Rural Development Administration, Republic of Korea. This research was also funded by the Foundation for Food and Agricultural Research (FFAR) grant number [NexGenHemp0000000016].

**Data Availability Statement:** The data presented in this study are available on request from the corresponding author. The data are not publicly available due to privacy restrictions.

**Acknowledgments:** Authors would like to acknowledge the NCSU Phytotron team (https://phytotron.ncsu.edu/) for their support and advise on this research project.

**Conflicts of Interest:** The authors declare no conflict of interest.

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
