# Peer review of "Flowering Response of Cannabis sativa L. ‘Suver Haze’ under Varying Daylength-Extension Light Intensities and Durations"

_horticulturae, doi:10.3390/horticulturae9050526_

Round 1

Reviewer 1 Report

The work is interesting for the knowledge of the inhibition of cannabis flowering.

It is necessary to attend to the following observations:

1. Check the spaces and punctuations in the manuscript

2. Order the figures as cited in the text.

3. Do not include figures of results in the materials and methods

4. In figure 1, indicate which is A and B

5. Indicating the characteristics of the water used to prepare the nutritive solution, as well as the elements that the nutritive solution contained, is important because the characteristics of the nutritive solution can influence the development of the seedlings.

6. Figure 6 is not necessary, it is irrelevant to include only one bar in the figure.

7. Do not repeat so many times DE and its meaning in titles of figures/tables, axis of the graphs, and title of the treatments, with the abbreviation it is enough.

8. Expand the discussion of the results and update the bibliography as much as possible, if necessary with bibliography of other crops.

Author Response

Response to reviewers

Manuscript:

Flowering response of Cannabis sativa L. ‘Suver Haze’ under varying daylength-extension light intensities and durations

We would like to thank the reviewers for their time and insightful feedback

Reviewer 1

Check the spaces and punctuations in the manuscript

Thank you very much for this comment; we have checked the punctuation and spaces.

Order the figures as cited in the text

Thank you very much; we have moved the figures to match the order of appearance in the text.

Do not include figures of results in the materials and methods

We have removed Figure 3 from the Materials and Methods section into the Result section.

In figure 1, indicate which is A and B

Thank you very much; we have added A and B to Figure 1.

Indicating the characteristics of the water used to prepare the nutritive solution, as well as the elements that the nutritive solution contained, is important because the characteristics of the nutritive solution can influence the development of the seedlings.

Thank you very much; the details were added under the Materials and Methods section (2.2 Plant growing conditions).

Figure 6 is not necessary; it is irrelevant to include only one bar in the figure.

Figure 6 was removed.

Do not repeat so many times DE and its meaning in titles of figures/tables, axis of the graphs, and title of the treatments, with the abbreviation it is enough.

For Table 1, Figure 3, and Figure 5. We would like to leave the definition and abbreviation since we believe every figure/table should be able to stand-alone (all explanations needed within the figure).

For Figure 2 and Figure 4, I agree with the comment, having both the abbreviation and the definition of DE is not necessary. We have removed the abbreviation.

Expand the discussion of the results and update the bibliography as much as possible, if necessary with bibliography of other crops.

Thank you very much for this comment; the current version of the manuscript was reviewed by three scientists in the field. Common feedback was to make the manuscript more focused on the responses of the specific cultivar to the treatments. We eliminated several areas of discussion based on those reviews. For the current version, I have added more discussion points in the manuscript (see track changes) using bibliography from field produced-cannabis. However, the additions were not extensive. We respectfully request to keep the current focus of the manuscript. Thank you very much for your consideration.

Reviewer 2 Report

In the current manuscript, Park et al. investigated the minimum light intensity and the required photoperiod to prevent incomplete inflorescence development. Two experiments were conducted: 1) to elucidate the required light intensity for day daylength-extension (DE), and 2) to identify the long-day photoperiod to prevent full and partial transitions to flowering. Two hypotheses were tested: 1) low light intensity (1-10 μmol m−2 s−1 photosynthetic photon flux density (PPFD)) is sufficient to prevent the transition from vegetative growth to the flowering of ‘Suver Haze’ (Experiment 1), and 2) ‘Suver Haze’ requires a photoperiod longer than 15-h to fully prevent the transition from vegetative growth to flowering. Although the topic is attractive, there are some concerns that should be addressed.

-Generally, the manuscript is well organized but has some typographical and grammatical errors.

-The paper title is well stated, and it is informative and concise.

-Abstract is well structured.

-The introduction was well written; however, it needs some improvements. The authors missed providing citations to support some sentences.

- Line 25: It is better, first, to introduce different applications of cannabis and then hemp and drug-type cannabis. So, I suggest the following sentences:

"Cannabis has been widely cultivated due to its industrial (DOI: 10.3906/bot-1907-15), ornamental (10.3390/plants11182383), nutritional (10.3390/plants11233330), medicinal, and recreational (10.1016/j.biotechadv.2022.108074) potentials. From regulatory and application perspectives, cannabis plants are categorized based on the level of Δ9-tetrahydrocannabinol (THC), one of the most important phytocannabinoids (10.1146/annurev-arplant-081519-040203). Plants are generally classified and regulated as industrial hemp if it contains less than 0.3 % THC in the dried flower (this level varies by country) or drug-type with more than this threshold (10.1016/j.indcrop.2020.113026)."

- Material and research methods are presented appropriately. The experimental setup and the description in the methods section are well structured, and the statistical analysis is correctly performed.

-The results obtained in this study are interesting. The discussion is presented correctly. However, the discussion should be improved, and previous studies should be discussed more.

- I suggest that the authors mention the limitations of the present study in the conclusion part and specify the follow-up tests.

Author Response

Manuscript:

Flowering response of Cannabis sativa L. ‘Suver Haze’ under varying daylength-extension light intensities and durations

We would like to thank the reviewers for their time and insightful feedback

Reviewer 2

Generally, the manuscript is well organized but has some typographical and grammatical errors.

Thank you very much for this comment; we have carefully addressed the issues.

The paper title is well stated, and it is informative and concise.

Thank you very much for this comment

Abstract is well structured.

Thank you very much for this comment

The introduction was well written; however, it needs some improvements. The authors missed providing citations to support some sentences. So, I suggest the following sentences:

"Cannabis has been widely cultivated due to its industrial (DOI: 10.3906/bot-1907-15), ornamental (10.3390/plants11182383), nutritional (10.3390/plants11233330), medicinal, and recreational

(10.1016/j.biotechadv.2022.108074) potentials. From regulatory and application perspectives, cannabis plants are categorized based on the level of Δ9-tetrahydrocannabinol (THC), one of the most important phytocannabinoids (10.1146/annurev-arplant- 081519-040203). Plants are generally classified and regulated as industrial hemp if it contains less than 0.3 % THC in the dried flower (this level varies by country) or drug-type with more than

this threshold (10.1016/j.indcrop.2020.113026)."

Thank you much, we have added the sentences and corresponding citations. The introduction is much improved.

Material and research methods are presented appropriately. The experimental setup and the description in the methods section are well structured, and the statistical analysis is correctly performed.

Thank you very much for this comment

The results obtained in this study are interesting. The discussion is presented correctly. However, the discussion should be improved, and previous studies should be discussed more.

Thank you very much for this comment. The current version of the manuscript was reviewed by three scientists in the field. Common feedback was to make the manuscript more focused on the responses of the specific cultivar to the treatments. We eliminated several areas of discussion based on those reviews. For the current version, I have added more discussion points in the manuscript (see track changes) using bibliography from field-produced cannabis. However, the additions were not extensive. We respectfully request to keep the current focus of the manuscript. Thank you very much for your consideration.

I suggest that the authors mention the limitations of the present study in the conclusion part and specify follow-up tests

Thank you very much for this comment; we have mentioned the limitations of the current research and the need for future research.

Reviewer 3 Report

The manuscript addresses results on Cannabis cultivation taking into account abiotic factors as experimental variables. I suggest making a Figure or Scheme showing how the data were collected (section 2.3), these could be showed on supplementary material.

Minnor corrections:

18-h and 21-h - Exclude the hyphen (lines 19-20, etc.)

Author Response

 Reviewer 3

18-h and 21-h - Exclude the hyphen (lines 19-20, etc.)

Thank you very much, we have removed hyphens

Author Response

Manuscript:

Flowering response of Cannabis sativa L. ‘Suver Haze’ under varying daylength-extension light intensities and durations

Congratulations on a well-organized, easy-to-understand and readable work. The innovative approach to work concerns the method of assessing the material consumption of the pistoncylinder unit of diesel engines in a biodiesel environment. The topic is very interesting, the purpose and implementation of the article are clear, and the information provided is informative. The contribution of the article is clear and it will certainly become the work of researchers in this field. This paper contains some technical issues that will need to be resolved before the manuscript is ready for publication.

This comment is not related to our manuscript, see highlighted text in comment.

Round 2

Reviewer 2 Report

All my comments have been addressed. The current version of the MS can be published in the journal.